# Reflections on Knowledge Production in Humanities from an Academic Exchange Experience

Mariángela Napoli

National Council of Scientific and Technical Research, Institute of Research in Educational Sciences, University of Buenos Aires, Puan 480, Anexo Bonifacio, 6° Piso, Ciudad de Buenos Aires 1406, Argentina; marar.napoli@gmail.com

**Abstract:** Over the last two decades, the knowledge production, research, and reconfiguration of universities have been understood as ways of giving new meanings to the university–society binomial. In this regard, humanities are the subject of multiple debates in the face of ideas about their impact in relation to the "other sciences". Based on these premises, this article sets out to explore possible meanings attributed by researchers to the concepts of commitment, mobilization, and transfer of research in humanities in view of the debates on the university–society interaction and the third mission of the university. The methodology used will address bibliographical analysis, theoretical background, and statements from different institutions, as well as the analysis of material from four interviews. As a first instance, the preliminary results show that strengthening critical thinking as forms of commitment emerge as central senses, focusing on Hungarian characteristics and productions in order to unravel the ways of understanding and imagining Eastern European reality. In this respect, the discussion of certain aspects of Western knowledge is seen as a task associated with social commitment with public universities as a focus of resistance.

**Keywords:** humanities; knowledge production; third mission of the university; knowledge mobilization

## 1. Introduction

The question of the definition of the humanities and the debates surrounding their status as knowledge production goes back a long way. The *studia humanitatis*, according to its appearance and use at the end of the 10th century, consolidated reading—and in a much more extended way language—as one of the places where humans have understood themselves and others [1] by analyzing their most decisively *human* ways of being and opening to the other. The study of these fields of knowledge was not entirely new, as rhetoric, dialectics, grammar, arithmetic, geometry, astronomy, and music, known collectively as the Liberal Arts [2], had already been cultivated in medieval universities for centuries under the names of the *trivium* and *quadrivium*. However, it was at the end of the Middle Ages that the syntagma *studia humanitatis* was used as a form related to the training of the orator, to the study of the everyday life of men, and again as that related to politics and civic education [3].

From a genealogical perspective, Michel Foucault (1968) already named "human sciences" as the knowledge that takes man as its empirical object. The human sciences did not appear until, under the effect of rationalism or some practical interest, it was decided to place man alongside scientific objects: they appeared the day when man was constituted in Western culture. For this, it was very necessary in these conditions that knowledge of man should appear, in its scientific direction, as a contemporary and of the same genre as biology, economics, and philology, following the model of empirical rationality. Consequently, for Foucault [4], the object of the humanities is not, then, language itself, but that being who, from within the language by which he is surrounded, represents to himself the meaning of the words he enunciates. In current discussions, the word humanities still allude to the

idea of *humanism* understood as a polysemous concept that is widely used to indicate a certain emphasis on human values, be they religious, scientific, or non-scientific values, which takes up this root.

From this perspective, the complexity of the epistemological status of the production of knowledge in the humanities is exposed, starting from the problem, as has been pointed out, of approaching it systematically as disciplines, as "sciences", or as fields of knowledge. Today, these debates continue to resonate through questioning on the academic–scientific map and take on special emphasis because inquiries into the meaning of human life are those that foreshadow that this knowledge is always revisited and allows us to enter into the debates on the production of scientific knowledge: the notions of accumulation of science, its validation, the registers of writing, all of them are deployed and established from the institutionalization (or attempt to) of the fields of study as disciplines [5]. Not only cognitive knowledge, but also aesthetic sensibility and moral sentiment, emotional empathy, and imaginative vision, along with many other kinds of intelligence and consciousness, are intrinsic to all that is known in the humanities. This means that knowledge in the humanities is contextual and relational, and therefore also historical and even personal [6].

From different traditions and interpretative perspectives, the humanities have become a field of study that develops in different academic spaces and produces, today, what we call research in the humanities. It is widely recognized that the humanities, as a field of knowledge, have always intervened in public debates. To this purpose, they have created institutional arcs for these debates to take root in the university and take up their propositional capacity associated with their concern to intervene in the public arena, in the social spaces of debate, and in the construction of social knowledge. This last idea constitutes the concept of intervention as a dimension that I intend to construct as an initial contribution in this article, based on the proposal to bring together notions from the field of study of the university and the role of knowledge production. The concept aims to allude to a materialization of knowledge that results in the production of concrete knowledge, but at the same time—as a novelty—it aims to be approached from the epistemological complexity of the production of knowledge in the humanities; this implies understanding how knowledge produced from the humanities can embody productions that are not only contemplative and/or interpretative of reality, but that allow incursions and approaches in different ways; in other words, knowledge that can produce, in light of the debates mentioned, a call to transform reality from research.

Therefore, if we assume that, from a close look at scientific and academic production, most of the production of knowledge and, therefore, the research produced, is carried out in universities as nodal spaces (within the possible nuances and numerous forms that this takes on), it is also central to propose a reflection on the current role of universities [7].

In his lecture *The university without condition (2022)*, the philosopher Jacques Derrida argued that the university should be recognized, in addition to what is called academic freedom, as an unconditional freedom of questioning and proposition (not only contemplation); from this point of view, the university should be thought of as a space for critical reflection [8]. It requires not only a principle of resistance but also a force of dissidence far removed from any utopian neutrality. The role it assigns to the humanities in it is major insofar as it deposits in it the capacity, after the notions of truth have been deconstructed within its framework, to exercise the profession of faith of a declarative and performative commitment.

In this regard, Eduardo Rinesi (2012), refers to the growing concern for what happens "outside" the university and for the modes of interaction between it and what remains outside. Specifically, this concern has guided a way of thinking about the academic question and the study of the humanities and social sciences, in particular [9]. The idea of an "outside" and an "inside" of the university implies a rethinking of the classic figure of the university as an "ivory tower" and its supposed disconnection with what happens outside. While the university has never been aliened to societal debates, and indeed today we can speak of a "third mission" [10] in relation to its links and interactions with society itself, it is important

to note the increasing attention that has been paid to notions of engagement, mobilization, and intervention as categories for analyzing academic and scientific productions. Currently, a vision persists that characterizes the university as something alien to society, as a separate institution, isolated from social problems; in this way, referring to the university–society or university–environment link is both a necessity and a criticism. To give an account of an orientation, it is the constant exercise of looking at what surrounds us and constructing ourselves as a part of it as a non-divided component; to think of the university as an institution in its society, and not as a matter outside of itself, questions its hermeticism and isolation and frames it within the very complex web of social relations conditioned and/or constituted by the culture, power, and ideologies of our time [11]. Consequently, it can be argued that there are as many concepts and definitions of participation or inclusion of the environment in the university as there are entities researching, writing, and debating about it. Attempts to define the concept and its component activities indicate contextual discrepancies and a lack of consensus among researchers, policymakers, universities, and funding agencies.

Now, if we accept this trend of reconformation of academic spaces, it is central to understand that the field of study of the humanities has been the subject of multiple debates in the face of ideas, always in tension, about its so-called uselessness and its impact compared to other fields. Since the 1960s, the idea of knowledge production has been in crisis, associated with a profound disinterest of humanistic studies in debates that lead to concrete ways of acting in the real world, as well as a position of self-enclosure [12] and hyper-specialization that led to a supposed exhaustion of its dynamics of knowledge production.

In this way, it is possible to visualize and establish that the forms acquired by the production of knowledge in the humanities in their particular dynamics have not been exempt from their participation in the current debates that currently affect the place attributed to the university and the role of the academic field in relation to the notions of linkage, productivism, relationship with social problems, critical perspectives and methodologies, efficiency, and forms of commitment and intervention. The central research question then becomes: What are the forms of mobilization and intervention of knowledge that the production of knowledge in the humanities acquires from the meanings given by the researchers themselves trained in different academic and institutional spaces? Likewise, other questions that arise are: What is the vision of researchers on the status of knowledge production in Humanities from the university–society debates, are there differences between the lines of research in humanities within the institutes/institutional spaces, what is understood by production and mobilization of knowledge that tends to enter and intervene in the real from the humanities, how is the contribution thought from the terms of theory and praxis for the humanities, how is the contribution from the terms of theory and praxis for the humanities? From what theoretical and political perspectives can socio-cultural processes and human life, in general, be thought of as a form of intervention in the real world from academic spaces?

Consequently, I consider it challenging to explore the possible meanings attributed to the humanities as fields of knowledge in their acquisition of possible relations with the "outside world" and the development of research products. It is novel to systematize the possible places given to its forms of producing knowledge from its links with society and the way in which these are interpreted and/or assumed by the academic field, through the visions of its own researchers. Thinking about this issue is—precisely—that which could allow us to return from inquiry and contemplation as a form of knowledge production based on the concerns of the present itself, the very thing that brings us back to our political community and its challenges for the future [1].

The task, then, that this article proposes, as a way of approaching the subject, lies in the description of the perspectives that emerge from the view that researchers and institutions construct on the place of scientific research in the humanities, as opposed to the debates that reflect on the dynamics of academic production with a focus on the real action that

research acquires; all of this, framed in the debates on higher education and the possible relations between universities–societies already mentioned.

## 2. Materials and Methods

The thesis is inscribed within the framework of interpretative sociology [13,14], recovering contributions from the phenomenological current [15] The theoretical framework allows coupling, on the one hand, the perceptions of the agents on their own practice without ignoring the structural elements contained in those practices. Following Zabala (2004), it is proposed to analyze the research topic/problem by combining the macro-social (in structural terms) and micro-social levels, where the emphasis is placed on the process of the production of meaning not only on their own practices but also on the meaning attributed to the structural elements and the perceived influence on their practices. Indeed, the theoretical framework allow us to overcome the agency–structure tension, present at the heart of the debates on the epistemology of knowledge [16].

The methodology of the proposed plan is mainly qualitative since it allows for describing the complex conceptual structures on which the practices, ideas, and beliefs supported by the actors investigated in the research are based, in this case, for research in the humanities. Thus, the first step will be to frame and systematize the theoretical production about the conceptualizations of the science–society relationship; the place of the university and academic spaces to produce knowledge, and, finally, the concept of mobilization and intervention framed in the debate on the production of knowledge in the humanities based on the meanings attributed by the researchers themselves. The bibliographic review and systematization will be useful, following Hernández Sampieri (2010) to detect key concepts, nourish the work with new conceptualizations, and deepen previous approaches to the problem [17].

The data collection techniques consist of semi-structured interviews (selected for the degree of freedom they allow), allowing the interviewee to elaborate on his/her arguments and allowing the interviewer to maintain a certain capacity to guide the interviewee, so that the interview does not result in the absence of structure and the conversation is directed to issues that are not central to the research problem to be solved [18]. The data analysis will be carried out with the techniques of content analysis and discourse analysis, guiding the study with grounded and data-based interpretation [19]. The data will be collected based on the definition of a theoretical sampling that is, at the same time, representative of the study problem.

For this article, I will present the analysis of material from four interviews and one questionnaire resulting from my exchange stay in Budapest, Hungary. The selection is based on the contacts I was able to make during my stay in Budapest through the tutors I had at ELTE; on the one hand, I tried to get in touch with an outstanding and recognized center of studies in the Humanities in the country, which could give me a global vision of the research perspectives of the field and the possible (or not) interest in the field of study of Science, Technology, and Society. For this purpose, I sent them an online questionnaire (via email) as they were not available to do it in person. In this way, a vision that encompassed researchers from different disciplines could serve as a starting point to move on to a micro-analysis. With this work developed, I sought to conduct interviews with trained researchers in different functions and roles: on the one hand, the sample involved a dialogue with a researcher who was involved in these issues and who also allowed me to have a vision associated with the production of knowledge from management; on the other hand, the interview focused on a professor with whom I took a course and whose work is associated with thinking about these issues but from a highly critical level and far removed from the management vision. Finally, the choice of a doctoral student allowed me to postulate another view to understand whether the new generations are debating these issues and what is their view on the forms of knowledge production in relation to the commitment and mobilization of knowledge. In this way, the selection allows the study to cover a wide spectrum and not to bias the proposed viewpoint. These four interviews were face-to-face

in the University and I was able to record and re-record them, with their consent and promise of anonymity.

Interviews conducted:

Interview _1. Trained researcher: Professor and PhD in History of Art
Interview _2. Trained researcher: Professor and PhD in Literature
Interview _3. Trainee researcher: PhD student in Literature
Interview _4. Trainee researcher Doctoral student in Social Anthropology
RCH Questionnaire

All questions are duly formulated in the body of the text and contain the raw material directly removed.

### 3. Conceptual Framework

Based on previous research [20–25] we have argued that there are multiple dynamics and modes of knowledge production within a field of knowledge, especially for the social and human sciences, according to the meanings that researchers (and research groups) give to their own practices. In conceptual terms, the aim is to recover for previous theoretical contributions on what we have called the third mission of the university [10]. and its multiple meanings under debate: extension, transfer, social commitment, and social impact of academic practices and ways of knowledge production. Specifically, the idea of "knowledge transfer" is a concept that has its origins in technology transfer and has mostly permeated all areas of science and technology management in the last forty years, but much more so in the last twenty. Both concepts did not arise to think about these practices in the social sciences and humanities, so that their appropriation continues to dispute their own meanings [26]. In this way, revisiting these notions based on the contributions and attributions of meanings from humanities researchers themselves is novel. Likewise, in this same sense, the concept of knowledge mobilization emerges for the discussion of research policies in the social and human sciences; that is, the requirement of knowledge production that implies going beyond its dissemination. It is worth mentioning that, in his research on this idea, Oscar Varsavsky had already developed the concept of politicized science to think about the production of scientific and academic knowledge, inspired by the debates that arose in Latin America between the 1950s and 1970s, which have been taken up again, in a more recent context, in an attempt to combine a science-oriented approach towards production and which proposes priority lines of work [27]. Other previous ideas are the approaches to the particular forms that knowledge assumes as added value or as a resource for the resolution of (social or public) problems. In general, the literature is eminently normative and prescribes desirable behaviors for a particular way of mobilizing knowledge.

With regard to the humanities, in particular, and their possible dynamics of knowledge construction in relation to the concepts outlined above, if, as Annick Louis (2022) points out, objects lose their power when they become conventionalized because they do not respond to current problems (i.e., they do not "act" in reality), the recovery of a possible renewed view of humanities research (and thus, of the scientific imagination) stands as a challenge. Studies on interdisciplinarity and transdisciplinarity become particularly relevant: the optimal outcome of an interdisciplinary humanities model is linked to engaged and expansive work that can help the humanities move beyond their stereotypical place in the academy as hyper-specialized and endogamous. It could also take us beyond counterproductive internal debates, which, to outsiders, seem meaningless, self-indulgent, and trivial [28]. Similarly, positions focused on reflecting on ways of understanding the mobilization and intervention of the humanities within the framework of the third mission argue that 20th-century theories were constituted as ways of transforming reality, of intervening in the world, of rethinking everything to make everything more dignified [29] In this way, the attributions of meaning to knowledge production in the humanities have been the subject of debates that have now become stronger and have been established as topics in universities. Numerous texts on research perspectives [1,5,28,30] show a self-reflexive

interest in one's own practices, their validity, their relation to the present, and to the "power" of critical action.

In previous papers and works (II Conference UNLAM about Humanities, VII Encounter of young researches in Education Science, IICE, UBA), I have highlighted, in a general way, the first debates around the intersection of issues on the role of the university and its relationship with research production and society, as well as the description of multiple meanings that derive from concepts such as commitment, intervention, and appropriation of knowledge, based on interviews with researchers in Argentina.

On the one hand, it is pointed out that the ways of writing and registering the work cannot be underestimated as a practice; therein lies a political positioning as a form of intervention, and this is not usually considered in research accreditation decisions. Nor are the lines of work thought of or highlighted in terms of contributions in themselves to understanding and contributing to critical thinking, which, it is argued, is a constituent part of the fields of knowledge themselves.

On the other hand, the senses of commitment, intervention, and social appropriation of the humanities are at stake: is it necessary to sustain what is committed and what is not from the humanities and for the humanities? Should we resign any hope that a humanistic production or knowledge can produce some kind of socially recognized value beyond the limits of institutionalized propositions, and what is the critical power of the humanities as a category to think about its place on the academic map? The first interviews left, in an exploratory way, a sense of unease insofar as few self-reflective channels on the production of humanistic knowledge were visualized, but with the explicit mention of a power of intervention in the public arena that has always existed and which, in the current context, is intended to be recovered as a relational and collective form of commitment.

### 3.1. Ideas about the Third Mission of the University Based on Declarations by European Organizations and Institutions

In this section, as a form of continuity with the interviews conducted with researchers from Argentina, the meanings and attributions of researchers obtained from my recent stay in Budapest, Hungary will be presented. The contribution of these perspectives allows us to add other visions from the European field as a general framework for approaching and thinking about similarities and differences with the Argentinean and Latin American fields within the debate on the third mission of the university. To this purpose, meanings were explored in relation to the place of the humanities in the production of knowledge and its possible links with the demands of society as a central axis of analysis, as a theme/problem that I am developing in my doctoral thesis. In order to achieve this objective, first of all, the analysis of sources and statements provided by interviewees from specific humanities institutions, as well as their own statements about these ideas, will be surveyed.

Declarations of European Institutes and International Organizations

As one of the first steps in my stay, the initial exchange with professors from different universities led me to investigate the institutional framework in which academic productions in Budapest, Hungary are developed. The first interview was conducted with a university lecturer and researcher trained in Art History and with an extensive academic and institutional management background in the national and international field.

His perspective, deeply centered on his role as a teacher, academic, researcher, and, above all, from his place in international institutions dedicated to thinking about the role of the humanities (and art history in particular) on the academic map in its links with society, was key to providing an overview of the perspective of analysis of the university–society debate that we wish to highlight in this article.

First, the interviewee highlighted three key conferences and meetings that have provided and continue to provide, today, a basic framework for thinking about humanistic research in the European context:

(1) The World Conference on Humanities: Challenges and Responsibilities for a Planet in Transition, held from 6 to 12 August 2017 in Liège, Belgium. It was co-organized by UNESCO and the International Council for Philosophy and Humanistic Studies (ICPHS).

(2) The European Conference on Humanities, 5–7 May 2021 in Lisbon, Portugal, jointly organized by the International Council for Philosophy and Humanistic Studies (ICPHS), the Portuguese Foundation for Science and Technology (FCT) and the Social and Human Sciences Sector of UNESCO.

(3) The Jena Declaration, carried out in 2021, from the incorporation of the objectives of the Sustainable Development Goals (SDGs) of the United Nations, which mobilized to think about scientific and academic policies worldwide and the role of the humanities and the arts in the current context.

About the first, in general terms, the central issue of the conference was to discuss the role of the humanities in a 21st century characterized by cultural diversity, the failure of different forms of single thinking, and the need to reincorporate the medium and long-term dimension into everyday reasoning. Already in a first preparatory instance, it was pointed out that:

> "Scientific knowledge, wisdom and human solidarity remain fundamental for human beings to face challenges that are not just problems, but complex dilemmas that require decisions based on citizen participation, peaceful coexistence and creativity, allowing everyone to believe in the possibility of a future characterized by equality and sustainability. In this context, the humanities have a historic role to play. They must remain a bulwark against xenophobia, intolerance, and fundamentalism. Their contributions should not remain in books, but be integrated into the knowledge of history, critical thinking and nuanced analyses of human ideas (. . .) Starting from a critical reflection on the disciplines of the humanities, i.e., languages and literature, history, philosophy and the arts, the role they can and should play in contemporary societies should be defined and reformulated, particularly in the context of the current crisis which, more than financial or economic, is in fact social, cultural and human".

> (https://unesdoc.unesco.org/ark:/48223/pf0000248002_spa, accessed on 15 February 2022)

From these ideas, it is possible to visualize a conception of the humanistic disciplines (understood in this way to think of them on the academic map) in their role as thought transfer and, above all, in their place of civic training; as expressed, the historical role of these fields of knowledge, already thought of as general knowledge that all society "must" acquire for decision-making, life in society and understanding the context in which one lives, gives rise to meanings that conceive of the humanities as transfers of values. However, it can be inferred that there is a need to "reformulate" their function on the academic map in order to be able to approach and intervene not only at the level of formal–theoretical discussion, but also through the production of knowledge that allows for the change of people's real life. In this way, a somewhat nodal and still unresolved epistemological problem is exposed: a certain traditionalist vision that "separates" the humanities and places them in a predominant place in order to train "good citizens" while, on the other hand, a need for renewal is "demanded" that does not only imply thinking of them as knowledge production in itself.

In this sense, it is also postulated that "the humanities, which can be called 'human sciences', pave the way for an education that addresses the complexity and interculturality that are underpinned as profoundly necessary for the state of the world today, using an approach based on a polycentrism that moves away from orientations largely centered on Europe and the West. The ultimate goal will be to rebuild the humanities on new foundations on all continents, framing the disciplines in a perspective guided by the world's cultural and linguistic diversity that responds to the challenges of society and

governance" (https://unesdoc.unesco.org/ark:/48223/pf0000248002_spa, accessed on 15 February 2022). At this point, it is central to highlight that the epistemological debate is taken up again to think of the humanities as disciplines/fields of knowledge, as well as the move away from certain forms or ways of producing knowledge to approach the challenges of organizing and planning scientific policies that contribute to rethinking the present. The premises indicated at the beginning of the article are therefore recovered, which, by way of hypotheses, are permeating the work of scientific knowledge, and a self-reflective work of the humanities in this context is urged.

Finally, it was stated by way of summary that a fundamental role of the humanities is precisely to strengthen this perspective from the academic field, while embracing a permanent interaction with all other sectors of knowledge and policies in society (https://www.cipsh. one/web/channel-112.htm, accessed on 20 March 2022). In this way, the constant mention of the need to continue to problematize the role of the humanities and their relationship with other "sciences" and from the interaction with society, its demands, and the specificities that contribute to the critical thinking that characterize these disciplines is visualized. The specificity of their contribution as a discipline is highlighted and characterized as crucial for illuminating other fields of knowledge.

As for the second conference, it is stated that it is "focused on the need to take more account of the human sciences in the development of public policy. The conference aims to put the human sciences back at the centre of scientific strategies and public policies to address contemporary challenges such as climate change and environmental degradation, migration, epidemics and gender issues, among others" (https://www.unesco.org/es/articles/conferencia-europea-de-humanidades-del-5-al-7-de-mayo-en-lisboa-organizada-por-portugal-y-la-unesco, accessed on 15 February 2022). Thus, in reference to the debate on universities–societies and the place of human and social sciences, it can be assumed that there is an interest in placing them at the center of scientific strategies and public policies to address contemporary challenges. If they are "key to solving current problems and, in close collaboration with other disciplines, can provide innovative answers and solutions, especially in the long term" (https://www.unesco.org/es/articles/conferencia-europea-de-humanidades-del-5-al-7-de-mayo-en-lisboa-organizada-por-portugal-y-la-unesco, accessed on 20 March 2022), it is necessary to rethink their epistemological particularities, their academic and scientific productions, and their specific place when it comes to producing knowledge that can dislocate their place "in crisis" and associated with the "utility" debate that produces misunderstandings in the era of efficiency. This complex place, then, which we intend to point out, allows us to visualize ways of renewing the humanities, but which do not produce a detriment to their critical capacity, a debate which in the preliminary interviews carried out in Argentina had already gained special relevance.

Finally, the 2012 Jena Declaration1 continued the work initiated at the conference "Humanities and Social Sciences for Sustainability", held in Jena (Germany) in October 2020. The conference was organized in collaboration with the Canadian and German UNESCO Commissions, the International Council for Philosophy and Humanistic Studies, the Social Sciences and Humanities Research Council of Canada, the World Academy of Arts and Sciences, the Club of Rome, the European Academy, and the International Geographical Union. The project is coordinated by the UNESCO Chair in Global Understanding of Sustainability at the University of Jena (Germany).

As a central aim, the focus was on "recognizing everyday practices as key drivers of transformation. This requires respecting the cultural, social, and regional diversity of these practices, as well as past experiences of adaptation. In this context, social sciences and humanities must play a central role in shaping sustainability policies" (https://www. thejenadeclaration.org/, accesses on 15 February 2022). To this end, three recommendations stand out:

- The arts in all their forms, along with the humanities and social sciences, are crucial to expanding mindsets and providing new perspectives on ways of life. This will enable

humanity to move from the age of extraction to cultures of regeneration, to achieve the SDGs with greater speed and depth, and to ensure measurable success.
- Establish universities and educational and research institutions as authentic examples for social transformation.
- Integrating the arts as well as findings from the humanities and social sciences in the co-design of future culturally and regionally diverse "sustainable livelihoods".

These latter ideas highlight, once again, the importance of giving centrality to the humanities, and particularly to the arts as a novelty that has also gained strength on the academic map, but which will not be addressed in this article. Closely linked to the Sustainable Development Goals, the social function of the University or what we have called the third mission of the University [10] is explicitly added as the central vehicle for transformation, in tune with the new senses of reorienting the production of knowledge and praxis in academic spaces. These objectives also permeated Latin American science policies, as frameworks for action, as can be seen in the Science and Technology Plan 2030 drawn up by the Ministry of Science, Technology, and Innovation of Argentina, which we have already analyzed [31].

It is worth noting, in terms of the interviewee's research activities, the realization of concrete proposals that he has indicated as ways of mobilizing and intervening from the production of knowledge. He emphasizes that he not only intends to carry out tasks as an art historian, critic, and writer, but also to actively participate in trying to transfer this knowledge to society; to this end, he mentions that "since its foundation in 2010, he has been working for Art Market Budapest—International Contemporary Art Fair" (interview N° 1).

Furthermore, his research project aims to make the viewpoint of contemporary aesthetics accessible to the Hungarian audience (through translations, edited volumes, and monographs); and to draw attention to the theoretical potentials of Environmental Aesthetics for the understanding of the interaction and dynamic relationship between man and his natural and built environment; to demonstrate the aesthetic aspects of actual theoretical and practical questions about the environment; and to build fertile relations with other disciplines, also triggered by the ecological crisis, such as (Cultural) Geography, Geology, Biology, Ecology, Landscape Architecture, Urban Studies, and the discourse on the Anthropocene, with which Aesthetics has hardly been connected before in Hungary.

In this way, there is an explicit need for the project to provide, first and foremost, concrete products that can be used to communicate research results more efficiently to society, such as translations. There is also a need to bring together theoretical and practical issues based on a specific problem: the environment and the human being in the contemporary world. In fact, the interviewee pointed out the following: "Our research, on the one hand, aims to reconstruct the tradition of Environmental Aesthetics that has been present throughout the history of Aesthetics, and, on the other hand, wants to raise aesthetic questions that are intimately linked to problems of culture and society today. (. . .) The central question of the research is what function the discipline of Aesthetics can have in today's society, when consumer societies in the most developed regions of the Earth have been irremediably replaced by a global risk society" (interview N° 1).

Therefore, with these documents pointed out by the interviewee and his academic journey as reflected in the interview based on the analysis of the objectives of his project, it is possible to advance in the meanings attributed to the places and/or debates in which the research development of the humanities in the European context is postulated in the first place. Secondly, taking the specific case of the interviewee, there is a clear mention of thinking about the impact, function, and mobilization of the knowledge produced to think about current issues concerning social problems. The implicit question seems to be guided by the notions we pointed out about rethinking the links between university, knowledge production, and society that postulates going beyond contemplation and allows us to solve—with the particular dynamics of the humanities, in this case, the history

of art—current social problems, rethinking issues of profound critical reflection, at the same time.

Likewise, the concern for the relationship of the humanities with other sciences begins to gain relevance from the question that arose from his project: "Does Aesthetics belong to the Humanities or to the Natural Sciences? A preliminary answer is that it belongs to neither, exclusively. Broadening the horizon could also allow us to rethink the place of Aesthetics within the general network of sciences, which can transform Aesthetics into a more interdisciplinary study and facilitate transdisciplinary cooperation with disciplines" (interview N° 1). Thus, the transdisciplinarity of the humanities emerges as a concept of renewing its epistemological status in terms of not thinking of itself as a marginalized "science" and as a way of understanding a broader contribution of the humanities in the specific current context.

*3.2. The Research Centre for the Humanities (RCH)*

Continuing on from the analysis of institutions in Budapest, Hungary, we will analyze the positions and meanings indicated by the Research Centre for the Humanities (RCH) to whom a closed questionnaire was sent and answered online. The RCH conducts essential research according to international standards in the fields of philosophy, literary studies, art history, ethnography, archaeology, history, musicology, archaeo-genomics, and classical philology. The nine institutes that make up the RCH examine and interpret the entire Hungarian past, reflecting on the challenges of the present. The fundamental task of the Research Centre is to explore the Hungarian cultural heritage and thereby strengthen the Hungarian identity. In this spirit, it establishes and maintains research groups at both central and institutional levels to study topics essential to national identity. Its mission is to bring specific historically accumulated Hungarian experiences into the international discourse, to renew communication with actors in the national and international scientific sphere, and to make humanities research visible to the public (https://abtk.hu/en/about, accessed on 20 April 2022).

In order to find out how they see it, the online questionnaire with the following questions and answers is transcribed:

General Topics in Humanities Research:

- Do you think that a relationship can be established between the Humanities and society, and how do you understand this concept?
  Yes. The fundamental task of the Research Centre is to explore and cultivate the Hungarian cultural heritage, and thus strengthen the Hungarian identity in society. In this spirit, the RCH establishes and maintains research groups at both central and institutional levels to study topics essential to national identity. Its mission is to bring historically accumulated Hungarian-specific experiences into international discourse, to renew communication with actors in the national and international scientific sphere, and to make humanities research visible to the public.

- How do you envisage the existence of mediations between academic theories and social debates? Are there any examples in the Humanities Research Centre?
  Politicians and government agencies often ask our researchers and experts for professional reference material on issues that require explicitly professional decision-making. Examples are questions concerning the conservation of monuments, the restoration of buildings or artifacts, national and international representation, archaeological excavations on construction sites or the decision on street names, to name but a few.

- Do you think it is important to emphasize humanistic studies in order to think about current social issues? Yes/no, why?
  Yes, the humanities can expose the cultural and historical aspects of social phenomena and developments, so recognising their importance in this field is essential.

  About the Humanities Research Centre:

- What are the main subjects of study in the Humanities at the Humanities Research Centre?

The Humanities Research Centre (RCH) conducts research essential to international standards in the fields of philosophy, literary studies, art history, ethnography, archaeology, history, musicology, archaeogenomics and classical philology.

- Are there themes that remain and others that could be considered emerging or contemporary trends? Yes/No. Which ones (if only I could name the ones that have increased in number over the last five years)?

  Yes, there are topics that remain as major turning points in national history or classic and canonized thinkers in our fields of research and others that could be considered emerging or contemporary trends. Examples of the latter are artificial intelligence as a moral issue in philosophy, women's history in history, archaeogenetics in archaeology, the networked locality of global change in ethnography, interactive visualizations of web data in literary studies, manuscriptibility in classical philology, digital archives in musicology, and so on to the cultural and historical aspects of the pandemic.

- Do these themes establish a relationship with critical thinking to reflect on what is happening in the present?

  Let us hope so. Critical thinking and knowledge of the major problems of the present that can be explained by the results of professional scientific research are very important for all societies around the world. The humanities can provide many examples of good or bad solutions from the past, so that today's decision-makers can learn from them to ensure better ways of dealing with the challenges of the future.

From the Institute's statements, it is possible to visualize a conception of the humanities as central to governmental and political decision-making, understanding the link that the center has with science policy sectors and the scientific knowledge that is "used" as a product of research.

Likewise, emerging issues, from the current complexity with which social relations and problematic situations of today's world are approached that call upon the "sciences" to intervene, (one can mention the advances in artificial intelligence, for example), appear linked to a space in which only the humanities can contribute. The idea of concentrating on a map of knowledge focused on Hungarian issues is also constantly highlighted, notions that I have visualized and will further specify, as a way of highlighting studies that tend to think about the country's present, a question that seems not to be sufficiently developed.

Finally, the idea of making the knowledge generated public stands out, another of the issues that concern the notions of communicability of science and that have permeated the humanities and social sciences in recent times, fundamentally in the era of the pandemic. Some of the proposals they have offered are https://mtabtk.videotorium. hu/ and https://open.spotify.com/show/2Ohx5ZgtevtqpnrBu2lJB1 (accessed on 20 May 2022) in which they communicate and debate through audiovisual productions and podcasts some of the achievements of their research.

Finally, future European congresses and centers were also mentioned by the interviewee as central to providing a framework for humanistic research in the European context.

HERA PROJECT is a network of European funding organizations dedicated to creating new opportunities for transnational and innovative research in the humanities. It aims to support a variety of activities, including research programs, conferences, workshops, and advocacy; HERA promotes the value of the humanities to society and to public policymaking.

Mention was also made of the Arts and Humanities Research Council (AHRC), a body charged with funding world-class independent research on topics ranging from philosophy and the creative industries to art conservation and product design. Its research addresses some of society's greatest challenges, such as modern slavery, exploring the ethical implications of artificial intelligence, and understanding what it is to be human today.

The XXI International Congress on New Trends in the Humanities, to be held from 28 to 30 June in Paris, will also be highlighted. It is based on the New Trends in Humanities Research Network, which provides a forum for communication with other people related to the same area of knowledge, exchanging ideas, and publishing their work. They seek to build an epistemic community where transdisciplinary, geographical, and cultural

relationships can be established. As a Research Network, they are defined by their thematic focus and the motivation to build action strategies determined by common themes.

## 4. Meanings around the Third Mission of the University in Humanities from the Point of View of European Researchers

Interviewee N° 2 belongs to the field of Literature, specifically, to the field of English Literature. First, it is important to highlight that, as far as we have been able to find out, university–society issues do not constitute a specific field, and no indicators or lines of work were found during the study visit, which aim to investigate this topic on the part of the researchers.

The full interview with questions and answers is transcribed below (own translation from English):

(1) First, I would like to know if you can tell me about your work with literary texts, the basis of your current project, what it is about and why you include the word *democracy* in it.

The project you allude to is a new project, led by a small research group at the Institute for Democracy at Central European University, which is still to be launched in September 2022. The title contains the mention of the classical term Utopia (from the 20th century) and the word Democracy, the idea being to think about how humanity is portrayed there. Particularly, I am interested in how freedom appears through the lens of social science and Hungarian dystopian texts. My interest is not only about Hungarian literature, but about the various studies that began to map social and political issues and recognised that democracy has been and is an important theme, from a critical stance in 20th century texts. Issues such as democracy or the idea of an elite that cannot rule the masses became relevant. In Western Europe, dystopias don't talk about democracy because they see it as a given, but in Eastern Europe or Russia, for example, democracy tends to be seen as very distant and problematic, at least questionable. It is very interesting how this is seen in Polish or Romanian literature. The idea is not to become a political philosopher, but I am interested in issues like autonomism, freedom of expression and the role of the media, how the media is structured today. For these, I consider transdisciplinary studies to be key, in the team there are political scientists, historians and political philosophers. The idea is to focus on the differences between Western utopian texts.

(2) How did you become interested in studying utopian and dystopian texts?

I've always been interested in literature, and I felt that you have to have a social approach and that's why these kinds of texts are really rich. There is a tension between the individual and the social, I think about Orwell being censored in the 1990s and that's when I said to myself: this is about us, about Hungarians. I read Anthony Burgess because I got Clockwork orange for Christmas, and I started to ask myself the question: Is it just entertainment fiction? There was practically nothing about it at university and it interested me; political and social issues didn't enter the university.

(3) How do you think these social issues and debates that interest you entered the university and how did they impact on the research?

The production of knowledge here was quite isolated for Social Sciences, it was not a common subject, they were more interested in narratology and classics. It was not very common. In the '90s everything was disappointing, here they wanted to be rich like in the West. Communism was over, but they weren't enthusiastic, they didn't see the way out. And even more so in the humanities, these topics were not common. Most teachers and researchers were quite apolitical.

(4) Has anything changed?

Social issues are not very common. They are not the focus of attention. Historians perhaps tend to be more interested, but not literature. It is something isolated, more like an ivory tower. Nowadays gender issues, feminist issues and so on are on the

agenda. These are issues for literature today. I think, of course, that women's issues are an important topic, as well as multicultural studies which are also on the agenda, but we must think about more popular culture studies (like cinema) and try to understand the impact on society. I think we need to revitalize the present: I don't just study just for the knowledge itself, I think there is a local culture that was formed in the 19th century, and this lack of debate tends to be reflected in the present and in people's understanding of what is going on. People prefer to be governed, I've noticed that and that's why it's important to address these topics.

(5) Regarding the terms that appear in the project: impact and public engagement, how do you understand these words, what meanings do you give to them?

The issue of engagement is a good one because I think this concept relates to the function of teaching, but also of involving the public. Education is very important. It is not about transferring one piece of information to another; it is something that happens and a group creates knowledge. The institute is trying to get more and more social engagement to organize open events, open to new forms of publications like podcasts. The university asks for publications, but it is not a public debate, it should be open to a wider public. They are not really interested in these ideas and that is not right. I am very happy that the foundation is thinking about podcasts, or videos that reach out to society. While the idea is not to become a media star, I do try to write in a more understandable way. In the British academy this has been a problem, to be more open, to communicate in a less indecipherable way. Regarding the term impact, I think from the humanities it's about how to encourage people to think and understand the complexities of the world. Public communication is often simplistic, and the humanities help to be able to deal with the complexity of issues. Very simple messages are always more attractive. I don't expect solutions to come, but by presenting the problems, I want students to understand what the problems are, I don't give answers, but I think they might find their own.

(6) Regarding the production of knowledge and the role of the humanities in this debate, what do you think is the role of the researcher, what should be the role of the university?

First, I was thinking that I often communicate in a complicated way, but the biggest problem is that in the media, thinking about ways of intervening in the debate, they are guided by the culture of celebrity. They are supposed to know everything, but they are not experts in anything. They talk about everything, and people listen to them. The problem with researchers is that they do not communicate in a simple way and on top of that, they are not very "attractive" like celebrities.

Then, regarding the influence on the university, although universities have prestige in other parts of Europe, here in Hungary they do not have prestige, it is very low. In a materialistic society like the one we live in where only money matters; it happens that research professors are very poorly paid and therefore not considered important people; this problematizes the impact of university professors on society. They are not respected. They are not recognised as important people. I already relate to this problem at my children's school.

I do believe that the impact we can make is through the students, yes, we can have an impact on the students, I think we can make them think, understand how to improve their mental capacities, and avoid accepting other opinions without criticizing them, without simplifying them, that's where we have the impact. The role remains in the teaching process. And here the humanities are central as they are taught more than anything else. There are some research areas, but they are not central in Hungary. This happens in other sciences, like natural sciences, but research is not very developed.

(7) The last question lies in the notion of social engagement for knowledge production from your perspective, how could you describe it?

I think teaching is still the main commitment. But in research I think it's a good question and it's more obvious in the social sciences than in the humanities. Translating,

for example, is a way of democratizing knowledge, which I also think is important. In other cases, such as my project, I feel it is a motivation to think and want to live in a democratic society and not in an authoritarian one, and that mobilizes me to think about decision-making processes. Thinking, writing, speaking serves to understand what others think. In the humanities, commitment does not appear directly. I also don't think you have to fall into the socialist notion, just focus on social issues like Marxism and think of other things as unimportant, even if it is not directly related. Here I would be careful in judging areas. Understanding how it was thought and how it is thought today (classical studies, for example) and the different perspectives is a way for things to change. And that allows us to see how they will change later.

From the transcribed questions and answers, various meanings and contributions to the above-mentioned concepts can be analyzed for the Hungarian academic map.

In the first place, it is possible to visualize a disconnection between certain subjects of study that have no place in the university, despite the framework and the strong emphasis in recent years on being able to link certain lines of research with current social demands and problems. The interviewee points out, on several occasions, the lack of interest in thinking about the present from the humanities in such a way that social issues enter as central axes, as well as in rethinking the so-called "third mission" of the university understood in its role of transformation and linking with society. Given the complex history and tense relationship between the Hungarian state (in its problematized democratic forms of government, as noted) and the university in its forms of knowledge production, it can be stated that the issues to be addressed are often financed by external organizations and centers, while the role of the university is maintained through its teaching activities. It is also stated that research is not sufficiently encouraged from these spaces, as well as the scarce support—in many ways, such as economic support—for the teaching task, which would give prestige and status to the figure of the teacher–researcher.

However, the notion of interdisciplinarity becomes important in terms of discussing the knowledge that the humanities possess, as was also analyzed in previous works as ways of revitalizing humanistic knowledge in order to have an impact on the present in the current context and research map; understanding social impact (an equivocal and polysemic category) as that which constitutes the transfer and formulation of diagnoses that serve as inputs for the formulation of efficient, medium- and long-term public policies. [23]. The idea of knowing for the sake of knowledge itself was discussed, although no concrete products or productions were identified that would allow us to go beyond this instance.

It also highlights that forms of impact or engagement in the humanities cannot be understood in terms of immediacy, an issue that has also been addressed in previous interviews given the epistemological modes, registers, and time that characterize this field of study.

In this way, the idea is consolidated that, although it is true that interdisciplinary approaches and dialogue are always a gain, they can be seen as the first step towards something even more fundamental and radical: the necessary dialogue between different paradigms with different ontologies, assuming the idea that this knowledge should not be postulated as passive custodians of specific knowledge and tradition, but as active constructors of change, knowledge, or interpretation [32].

Regarding the meanings of commitment, as analyzed in previous texts [10], it is pointed out that commitment cannot be defined beforehand or based on the choice of specific topics. In other words, the quality of a production does not consist of commitment, although quality is attached to commitment, since with it or without it, it becomes a mere allegory of the author's subjective intention.

Underlying the idea, then, as we have pointed out [33] that commitment is a relative and contingent magnitude to those who intervene in the process of its own definition, the interviewee points to teaching as the greatest commitment and as a source of training for young students and/or researchers in training that will result in their ways of developing in society. In this way, his conception can be associated with the ideas of humanistic and

artistic knowledge production as a practice of resistance, because it opens up the horizon of the possible and rehearses modes of existence of what we can become [34]; in this way, his conception is nested in the idea that humanistic knowledge production implies discussing the definitions of the state of things and the very nature of things: its value is nested in a potential for problematizing life as a whole, as a result of human practice that allows it to be visualized as a form of intervention in the public arena.

It also revives the debate that has taken place throughout the history of humanity, from the identification of the *humanist* as an individual belonging to a cultured elite and prone to abstractions and theoretical elaborations, to the figure of the committed intellectual who must reveal his ideology and actively participate in political tasks or social movements or, at least, try not to remain on the sidelines of the urgencies of his time.

On the other hand, the interviewee made no mention of ways of linking humanistic knowledge as a basis for thinking about public policy as visualized in the congresses analyzed, as one of the functions that knowledge in these fields could adopt to generate impact, but he did point to the growing interest in the democratization of knowledge and the notions of public communication of science. Recent technological changes that make it possible to generate ways of disseminating or "making" the production of knowledge closer to society—without simplifying it, another notion of great interest that he mentioned—also became important as efforts to do so are envisaged from the creation of social networking tools or internet circulation, as the RHC also does. In this way, the interviewee recaptures the notions of the third mission of the university that are pointed out as part of the obligations of universities to communicate "to the public" the research that is carried out. In this way, the action of translating scientific terminology into everyday language, where necessary, becomes relevant and valued [10].

Finally, teaching is again highlighted as the clearest form of intervention for the interviewee, which is located, following the table developed in our text, as the first dimension of activities of the third mission, which is the training of professionals. From this perspective, the act of teaching is recognized as a central axis to focus on experience, resignification, and attribution of meaning to things in the world and to oneself, which allows the subject to organize reality and self-realize, situate themselves, and participate in a given space and time [35]. These notions are keys that allow us to think about the epistemological status attributed, today, to the humanities to address another dimension of analysis that would make it possible to link this knowledge with concrete ways of intervening. This makes it possible to argue that, without a debate on its specificity in terms of knowledge production, it is difficult to think about ways of mobilizing this knowledge.

Now, to close the interview stage, the meanings attributed to these themes by two researchers in training will be pointed out, to give another vision from the field of young researchers and which allows us to think about the future of the ideas set out above. The students are studying for their PhDs in Literature and Anthropology, respectively, and come from the field of political science. First, about the forms of knowledge production, they both pointed out the following: "For me, interdisciplinarity is natural" (interview N° 3 and N° 4). In this way, the conversation began, thinking about their respective doctoral work, from the idea that the humanities should be thought of from a non-isolated place and in contact with "other sciences". To this end, they also pointed out: "I believe that people should try to think outside their boxes for some of their subjects, obviously not all of them. Interdisciplinary studies have never been encouraged in universities. It's a bit of a gamble. So, if the university wants to make sure that students graduate, that they do well, I tend to think let them stay in their fields. I'm not devaluing any other kind of research, but I do want to say that it's more difficult and complex to do this kind of research. And besides, you end up missing crucial points that are later refuted in peer reviews and other forms of evaluation" (interview N° 3).

In the same way, the interviewed N° 3 pointed out that: "You have to be a specialist in a very narrow niche. And the same is reflected in the academic world. The humanities are basically in crisis because people won't pay for them. Today everything is driven by

money" (interview 3). Therefore, the importance of including, when reflecting on academic production, the productivist ethos against which interviewee N°. 2 also pointed out, as a constant concern to postulate the importance of promoting, from academic and scientific policy bodies, research in the humanities and with them the evolution of the evaluative culture that does not allow the epistemological specificities that were pointed out to be confronted, is highlighted. It seems that the disciplinary logic still weighs heavily on the notions of relationships between fields of knowledge, as well as the idea that the humanities still seem to be stuck in "niches" that further accentuate the notions of the "ivory tower" that emerged in the interviews as archaic but still valid ways of thinking about the production of knowledge away from what happens outside.

Regarding the notion of impact, the following was stated: "Maybe compared to other sciences, the social sciences and humanities we don't have the direct impact of making a drug or trying to beat COVID with vaccines and stuff. But maybe just to think about going beyond contemplation, what we can do is to establish a link of contact between what is going on out there. Between the university and what is happening in the outside world" (interview N° 3). In this way, the need to produce knowledge in such a way as to allow a renewed search for thinking about the current issues that society demands and to be able to act and intervene. The term linkage mentioned allows us to think again about a search for contact with that "outside" from the University that Rinesi (2012) points out, but understanding the mediations that constitute humanistic knowledge and its specificities that are not equally evaluated or understood in the same way as other sciences when thinking about scientific policies or forms of interaction between the "outside world" and the research field of these areas [9].

Finally, interviewed N° 3 posited that: "Studies that are so encapsulated in themselves that they have no practical relation to other kinds of knowledge, in that sense they are useless. But I also think that they can generate very interesting and positive theoretical things, as knowledge by itself defeats its own purpose.I think interdisciplinarity can not only revive the humanities, but also enrich other disciplines. If more interdisciplinary research was done, it would really encourage more" (interview N° 3).

In this way, we can see the importance of the humanities opening to other fields and in relation to thinking about their contributions to other knowledge and *vice-versa*, but without neglecting the importance of generating theoretical and reflexive knowledge, which is a constitutive part of their epistemological status. The ways of fostering and circulating humanistic knowledge seem to be linked to its links with another knowledge.

On the other hand, interviewee N° 4, anchored in her anthropological work, showed that her work involves other forms of impact different from that of the literary field. To demonstrate this, from her concrete experience she stated the following: "the people with whom we were linked to think about fieldwork took our advice and began to work in a different way, our work modified community practice, we helped them to think about how to build correctly from their own interests" (interview N° 4). In this way, a linking style associated with social commitment is visualized [26]. This style states that "this style is characterized by manifesting a motivation linked to the commitment to solve or collaborate with the solution of a local social problem (. . .) They make explicit the search for solutions to problems caused by inequality, poverty, oriented towards the common good of the collective that is broader than the specific target object that they describe as such" [26], p. 94. At the same time, she pointed out that the volunteering experience that was part of their fieldwork allowed them to understand and question, for example, "whether public spaces really belong to those people", in relation to the cultural institution they took as a case study. Consequently, contact and engagement with these groups also enriched the researchers who, as they also pointed out, "we understood that our form of social engagement had to do with forgetting that this was academic research, we got involved" (interview N° 4). The interesting topic about this statement, as noted above, is that it reinforces the idea of understanding different forms of commitment based on the different logics of the humanistic fields of knowledge; social anthropology allows for other dynamics

that are more clearly visualized in its projects and concretions in terms of interaction and appropriation by society and exposes the construction of meanings of this concept that is attributed differently from each work team and from each discipline. Finally, regarding the ways of writing and circulating knowledge, it was also stated that "university language tends to be a privilege, here we try to write it together, to be more involved in their ideas, it is a way of decolonising the academy" (interview N° 4). Thus, the need and interest in continuing to rethink the university–environment binomial is exposed.

## 5. Results

Within the framework of the aims outlined above regarding a possible reflection on the current epistemological status of the humanities, their role, and their links with a renewed view of the university, both in Latin America and on the basis of the statements made by European organizations, institutions, and centers, they express some of the following ideas: (1) the knowledge production in humanities is constituted as a driver of significant contributions to society from renewed readings of reality tending to contribute to a deeper understanding of the socio-political conjuncture; (2) these renewed forms are closely related to the meanings that interdisciplinarity can contribute, from contact with other social knowledge; (3) the forms of knowledge production in the humanities maintain lines of work that have registers, methodologies, and times that are different from other "sciences", which must also be valued by evaluation, through scientific and academic policies that do not take into account their specificity. Likewise, regarding the senses of commitment, social appropriation, and mobilization of humanistic knowledge encompassed in the different aspects of analysis of the third mission of the university, it is pointed out that humanistic knowledge allows one to not only provoke a theoretical reflection that enters public debate (as has been profoundly highlighted in the pandemic) but also to intervene in the modification of the status of reality. Forms of social communication of science, as well as functions of education and the transmission of knowledge, were highlighted as central. Many of the projects and products mentioned aim to contribute to the transformation of concrete problems associated with current realities that they wish to change, although there are substantial differences between disciplines. It should be noted that the notion of social commitment, problematic as a relational term and as a conjunctural and epochal construct, allows us to visualize how in the specific situation of Budapest, Hungary, for example, through the study of democratic forms, it allows us to affirm that humanistic knowledge contributes to its development and constitutes an inescapable focus. Strengthening critical thinking as forms of commitment emerge as central senses in most of the interviewees and in the centers of production, focusing on Hungarian characteristics and productions to unravel the ways of understanding and imagining the Hungarian and Eastern European reality. In this way, the discussion of certain Western knowledge is seen as a task associated with a social commitment to think about the complex Hungarian present, with the public university being a focus of resistance for debating these ideas.

## 6. Discussion

The positions on the values that the humanities encompass—from different points of view, when thinking about their place in the production of knowledge and their importance in the academic map expose historical debates that have been discussed from the history of the humanities. This question allows us to address the importance of research from inquiry and contemplation as forms of knowledge production based on the concerns of the present itself, which brings us—the researchers—back to our political community and its future challenges [1]. From this perspective, it would be possible to address key issues about subjects and questions about humanity in different ways that contribute to the construction of knowledge from a place where contemplation and thought lead to different forms of intervention, as well as to (self) reflect on the role and place of humanistic research in the scientific map and its link with society. This concept allows us to understand how the humanities and the academic community are currently participating in the debates within

the framework of studies on the university, science, technology, and society, topics that have usually been addressed by the so-called "hard sciences".

Regarding the production of knowledge in the humanities, there is still a strong debate about its futility/uselessness [10,32]. Positions focused on reflecting on the ways of understanding the intervention in the present of the humanities argue that the theories of the twentieth century are ways of transforming reality, of intervening in the world, of rethinking everything to make everything fairer [29]. Various works and congresses on research perspectives in humanities from different approaches [1,5,30,36] allow us to argue that the production on the subject is relevant and current. However, in general terms, no progress has been made in the development of deepening works that allow us to hierarchize the relationship between the humanities and their environments, or that recognize the particularities of knowledge production in terms of its mobilization and intervention capacities [21]; essays and approximate formulas continue to be developed especially from management offices, but it remains an unresolved issue and problem [10]. Consequently, as a research problem proposal, it is challenging and less addressed to venture into the aforementioned discussions to which the humanities have also contributed and continue to contribute in a specific way. Given their vacancy, it is novel to systematize the possible meanings to their knowledge production, starting from the concepts of the third mission in the university to account for the ways and forms in which these are interpreted and/or assumed by the researchers themselves in their research practices. The study takes up the themes on universities–societies, which have been named in many ways, such as extension or transfer, mission, social function, social commitment, and the concept of intervention, among others. Indeed, it is assumed that there are processes of production of humanistic knowledge that become significant as a contribution to the development of society and, therefore, these phenomena are plausible to be analyzed [23].

Consequently, as a preliminary result of the interviews presented to the researchers trained in the field of humanities studies from different academic and institutional spaces in Budapest—in the international European framework—it can be stated that they interpret the notions of the third mission of the university in different ways. Indeed, the realization of different research practices differentiates them from other sciences and allows them to visualize specific forms of knowledge production.

## 7. Conclusions

As a summary, this paper aims to expose a topic that has not been sufficiently explored to think about the links between knowledge production in the humanities and its links with society. To this end, firstly, a review was made of the theoretical framework and the state of the subject and the methodological proposal of the work was given. Then, interviews with researchers and a questionnaire carried out in a particular institutional space were presented, with the aim of interpreting the meanings that emerge from the academic community's own visions on these issues. Their analysis allows us to understand that the academic practices of humanities researchers differ epistemologically from other sciences and constitute their own methodological approaches to the social phenomena they study. In this way, interdisciplinary work, for example, constitutes a central form that must be recovered, as well as the teaching function to contribute to critical thinking, which continues to be the first axis of transmission and contribution of knowledge.

**Funding:** This research received no external funding.

**Informed Consent Statement:** Informed consent was obtained from all subjects involved in the study.

**Data Availability Statement:** The raw data supporting the conclusions of this article will be made available by the author on request.

**Conflicts of Interest:** The author declares no conflicts of interest.

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
