# Peer review of "Reflections on Knowledge Production in Humanities from an Academic Exchange Experience"

_knowledge, doi:10.3390/knowledge4020011_

Round 1

Reviewer 1 Report

Comments and Suggestions for Authors

Production at universities in the sense of the humanities is certainly a very interesting topic. The authors try to briefly deal with the given topic. It is not only a theoretical treatise, but also a monitoring of empirical indicators. In the introduction, the authors reflect on the antiquity of the university as an institution. They are also aligned with Foucault's concept of the humanities. They remind us that these sciences are not only about knowledge, but also about values. The authors recall that Jacques Derrida said that at the university we can freely question and propose new ideas. They also think about the outputs of universities and their ideas towards the external environment. The selected paragraphs are reflections on production in the humanities. In the intentions of the conceptual framework, the authors quote and paraphrase several important authors. The authors mentioned the relationship between science and politics, as well as the various influences of the humanities on the social field. Empirical data come from the regions of Argentina and Hungary. The authors reflect important humanistic reflections. The humanities prepare the ground for education. According to the authors, the humanities need to be returned to the center of scientific strategies. For spreading new perspectives on ways of life. They also think about the tasks of aesthetics. They are also considering a research center for the humanities. They represent a questionnaire with questions for respondents. The authors present the entire interview. They return again to the question of the apolitical nature of science. The discussion is relatively short, but to the point.

It is necessary to formulate a conclusion. A scientific article must not be without a conclusion. It is also necessary to specify and unify in the form of bibliographical references.

Author Response

Dear reviewer, I have reviewed the file and made the appropriate modifications.
Best regards

Reviewer 2 Report

Comments and Suggestions for Authors

The article is devoted to the analysis of peculiarities of knowledge production in humanity scienences.

However, the unconventional structure and the lack of a lot of expected information makes it very hard to analyse the scientific contribution and impact of the article.

I strongly recommend structuring the article according to the widely-used IMRAD model, in particular:

1. Introduction that should include review of related works and clearly formulated research questions ( RQ).  Formulating research questions at the beginnig of the study will allow the readers to assess the study's goals and see how well they are achieved.

2. Methods which describes step-by-step the way the study was conducted (including the questions used in interviews) and provide references to the methods used for analyzing the results. This section must also contain description of your sample (e.g., how the people for the interviews were selected). Currently, the article has a small section on methods near the end which does not provide sufficient information and a step-by-step procedure.

3. Results which describes the results obtained by the procedure described in the previous sections.

4. Critical Discussion of the results, including comparing your findings with findings of other researchers and discussing threats to the study's validity.

5. Conclusion, briefly summarizing the study outcomes.

The article also has technical errors:

1. Reference format does not meet the journal's guidelines

2. There is a reference in the abstract which is normally avoided.

3. There are many references from "the team of which I am part" without clearly describing how each of them is relevant for the research, which can be considered inappropriate citations.

4.  There's a strange non-informative part at the end of the article titled "Interviews conducted"which does not contain real information. Providing the real raw content of the interviews would be beneficial in the form of additional supporting materials.

5. There are no required sections at the end of the article (i.e., sections on ethical approval, informed consent, funding, etc.)

Author Response

(The authors gave the same response as above.)

Round 2

Reviewer 2 Report

Comments and Suggestions for Authors

Dear author.

According to the Instructions for Authors (https://www.mdpi.com/journal/knowledge/instructions , section "Editorial Process and Peer Review") you should "provide a point by point response" to the reviewers' comments. Instead, you (possibly by mistake) attached a copy of the article in the "authors reponse" field.

I organized my comments in the first review into a numbered list for your convenience. Please, provide the necessary response which kind of changes were done to the article and the numbers of lines where those changes can be found. It would be good to highlight the changed parts of the text too, but it is not necessary. 

Author Response

Dear reviewer,

The changes I have made are the following:
-I completely rearranged the section on Materials and Methods as completed and detailed as possible (l. 118-161)
-I expanded the following sections: results (l. 646-668), discussion (l. 670-701) and conclusion (l. 702-711).
-I specified and unified in the form of bibliographical references (l. 717-803)

-I added the required sections that were missing (l. 713-715)

-I highlighted all te changed parts

Round 3

Reviewer 2 Report

Comments and Suggestions for Authors

The manuscript's structure was significantly improved and now meets the accepted standards of a scientific publication.

There are technical issues that can be corrected, for example:

1. the list of interviews at the end of chapter 4 can be made a table

2. long URLs in the body of the manuscript can be made references or footnotes

3.  Sections 2.1, 2.2. and so on are in fact subsections of the Section 3, not 2

4.  the numbered list in Section 4 should be formatted as a list, with each element starting in its own line of text

5.  The article has a strange note at the end "he statements, opinions and data contained in all publications are solely those of the individual 164 author(s) and contributor(s)..." - does it really have 164 contributors?

6.  You may want to elaborate on "Various works and congresses on research perspectives in humanities from different approaches (Culler, 1997; Ciordia et al. 2011; Benneworth, 2015; Louis, 2022) allow us to argue that the production on the subject is relevant and current" - so that it wouldn't look that your only argument is that someone wrote that it's relevant and current. What in those works do you see as better argumented than in the critical works mentioned before?

Comments on the Quality of English Language

There are minor language issues in the manuscript. E.g., "expose historical debates that have been discused from the history of the humanities"

"there is an imaginary about its futility/uselessness" (using "imaginary" as a noun)

and so on.

Author Response

1.I don´t know if I understood the note but I made a table with the list of interviews in chapter 2, methodology.

2. l have made the necessary footnotes.

3.  I rearrange the  sections.

4.  I noticed that the questions are lined up.

5.  I corrected the mistake.

6.  elaborated the concept from LINE 681.
